# Arcuate Fasciculus Microstructure Predicts Alcohol Dependence Risk through Higher IQ

**DOI:** 10.3390/brainsci13010129

**Published:** 2023-01-12

**Authors:** Toshikazu Ikuta, Paige B. Kessler, Alexandria M. Swoboda, Amy K. Fisher

**Affiliations:** 1Department of Communication Sciences and Disorders, University of Mississippi, Oxford, MS 38677, USA; 2Department of Social Work, University of Mississippi, Oxford, MS 38677, USA

**Keywords:** arcuate fasciculus, IQ, alcohol dependence risk, diffusion tensor imaging, tractography

## Abstract

IQ has been found to correlate with alcohol consumption, with a higher IQ being a risk for alcohol misuse. Furthermore, recent research has shown that the microstructure of the arcuate fasciculus is associated with IQ. This study therefore aimed to examine the association between the arcuate fasciculus microstructure, IQ, and alcohol dependence risk. In this study, we performed probabilistic tractography between Wernicke’s and Broca’s areas in the left and right hemispheres to examine the association of the arcuate fasciculus’s integrity with IQ and alcohol dependence risk, using DTI data from 344 individuals. Data regarding IQ were obtained from the Wechsler Abbreviated Scale of Intelligence (WASI-II). Alcohol substance involvement (SI) score was derived using the National Institute on Drug Abuse (NIDA) Quick Screen and was used as an index for alcohol dependence risk. Both the left arcuate fasciculus and IQ were found to have a significant association with alcohol dependence risk. A mediation analysis revealed that this association between the left arcuate fasciculus microstructure and an alcohol dependence risk was mediated by IQ. It is suggested that the left arcuate fasciculus microstructure is associated with IQ which is associated with alcohol dependence risk. While alcohol consumption is known to be robustly toxic to the brain, the left arcuate fasciculus shows exceptional characteristics in which its microstructure integrity is positively associated with an alcohol dependence risk through higher IQ. Clinical implications are discussed.

## 1. Introduction

The microstructure of the arcuate fasciculus (AF) was recently shown to be associated with IQ (Ikuta et al., 2019). The arcuate fasciculus bridges two distant structures: Wernicke’s area in the temporal lobe and Broca’s area in the frontal lobe. This structure has been shown to be highly elaborate in humans as compared to non-human primates [1]. Therefore, the AF may represent intelligence unique to humans.

IQ has been found to be associated with alcohol use. For example, a study of 99 countries showed that a higher IQ predicts a greater level of alcohol consumption [2]. A longitudinal study over 37 years has shown that a higher IQ in adolescence is a risk for alcohol-related hospitalization [3]. IQ has also been shown to be associated with a preference for wine over other beverages [4]. These results indicate that higher IQ is a risk for alcohol misuse.

At the same time, it should be noted that the toxicity of alcohol consumption has been well established, and that the influence of this toxicity is known to be relatively widespread across the brain [5]. Additionally, higher levels of alcohol consumption and alcoholism have been found to result in a reduction of cortical gray matter [6,7,8] and white matter [9,10,11], as well as in subcortical structures extending to the brain stem and cerebellum [12,13]. Diffusion tensor imaging (DTI) studies found reduced fractional anisotropy in alcoholism, suggesting damaged myelination in the white matter [14,15]. Such research demonstrates that alcohol misuse has negative effects on the brain.

Despite the fact that alcohol has been found to cause harm to the brain, associations have been independently found between both the AF & IQ as well as between alcohol use and IQ. In this study, we aimed to examine this triangular relationship between the AF, alcohol use, and IQ. Using diffusion tensor imaging data and probabilistic tractography of the AF, we examined whether the AF microstructure is associated with alcohol addiction as well as IQ.

## 2. Materials and Methods

The AF data used in this study followed our previous study [16]. Briefly, the Nathan Kline Institute-Rockland Sample (NKI-RS: http://fcon_1000.projects.nitrc.org/indi/enhanced/) [17] data were obtained from the Collaborative Informatics and Neuroimaging Suite (https://coins.trendscenter.org/). From the database, data were obtained from both the Alcohol, Smoking, and Substance Involvement Screening Test (ASSIST), which is a self-administered questionnaire, and the Wechsler Abbreviated Scale of Intelligence (WASI-II) including a full-scale IQ composite score. The alcohol substance involvement (SI) score was used as an index of alcohol dependence risk.

The DTI series consisted of 128 volumes of noncolinear directions as well as nine volumes without diffusion weighting (TR = 2400 ms, TE = 85 ms, matrix = 128 × 128, FOV = 256 mm, b ≈ 1500 (except the first volume b = 0)). Each volume consisted of 64 contiguous 2-mm slices with 2mm^3^ isotropic resolution.

Image data were processed using the Functional Magnetic Resonance Imaging of the Brain Software Library (FSL version 4.1.8; Oxford, UK; http://fsl.fmrib.ox.ac.uk/fsl). Eddy-current-induced distortions were corrected and head-motion displacements were corrected through affine registration of the 128 diffusion volumes to the first b = 0 volume using FSL’s Linear Registration Tool. The b-vector table (i.e., gradient directions) for each participant was then adjusted according to the rotation parameters of these corrections. Non-brain tissue was removed using FSL’s Brain Extraction Tool. Fractional anisotropy (FA), an index of white matter integrity, was calculated at each voxel of the brain by fitting a diffusion tensor model to the raw diffusion data along with using weighted least squares in FSL’s Diffusion Toolbox.

The white matter tract between Broca’s and Wernicke’s area was isolated as the AF using probabilistic tractography as described in our previous study [16]. The script to conduct probabilistic tractography is also available online (https://olemiss.edu/projects/dnl/codes.html). Briefly, tractography was conducted within the white matter of each hemisphere above MNI *Z* = 0 by having two seed regions: Broca’s area (the pars triangularis and pars opercularis) and Wernicke’s area (the posterior superior temporal gyrus and planum temporale) defined in the the HarvardOxford cortical atlas [18]. The mean FA within the left and right AF (thresholded at a normalized probability value of 0.04) was calculated for each individual. In 334 individuals, (42.99 ± 22.78 range 16 to 85 years old, 113 males and 221 females; Table 1), the Arcuate Fasciculus FA (AFFA), IQ, and National Institute of Drug Abuse addiction data were available. The mean AF is shown in Figure 1.

Statistical analyses were conducted independently for left and right AFFA. A multiple linear regression was calculated to predict alcohol dependence risk based on AFFA and age. Another linear regression was tested to predict alcohol dependence risk based on IQ. In order to test whether the relationship between the AFFA and alcohol dependence risk was mediated by IQ, a mediation analysis was conducted using the *MBESS* package [19] on R 3.6.1. In the mediation analysis, the residual from a regression to predict AFFA based on age was tested to examine the age-independent association between AFFA and alcohol dependence risk.

## 3. Results

The left AFFA was associated with alcohol dependence risk (Figure 2). In a multiple regression to predict alcohol dependence risk based on AFFA and age, a significant regression equation was found (*F*(2,321) = 5.813, *p* = 0.0033) with an *R*^2^ of 0.29 in the left AFFA. The predicted alcohol dependence risk is −3.18 + 26.02 (left AFFA) + 0.067 (age). The left AFFA (*p* = 0.028) and age (*p* = 0.0044) significantly predicted alcohol dependence risk. However, the right AFFA did not show a significant association with alcohol dependence risk.

In a linear regression to predict IQ based on age-adjusted left AFFA, a significant regression equation was found (*F*(1,322) = 8.53, *p* = 0.0037) with an *R*^2^ of 0.023 (Figure 3).

IQ significantly predicted alcohol dependence risk. In a linear regression, a significant regression equation was found (*F*(1,322) = 10.76, *p* = 0.0012) with an *R*^2^ of 0.29.

The relationship between the left AFFA and alcohol dependence risk was mediated by IQ (Figure 4). The standardized regression coefficient between the left AFFA and IQ was statistically significant, as was the standardized regression coefficient between IQ and alcohol dependence risk. The standardized indirect effect was (0.16) × (0.16) = 0.026. We tested the significance of this indirect effect using bootstrapping procedures. Unstandardized indirect effects and partially standardized indirect effects were computed for each of 10,000 bootstrapped samples, and the 95% confidence interval was computed by determining the indirect effects at the 2.5th and 97.5th percentiles. The bootstrapped partially standardized indirect effect was 0.65, and the 95% confidence interval ranged from 0.12 to 1.41. Thus, the indirect effect was statistically significant.

## 4. Discussion

This study examined the association between the arcuate fasciculus and alcohol dependence risk. Despite findings that alcohol consumption is harmful to the brain, alcohol dependence risk was positively associated with the microstructure of the left arcuate fasciculus (left AFFA). It is known that the arcuate fasciculus is positively associated with IQ and that IQ is positively associated with increased consumption of alcohol. We found that IQ mediates the positive association between the left AFFA and alcohol dependence risk. This suggests that stronger AF connectivity increases the risk for alcohol dependence through higher IQ.

Our current finding is primarily inconsistent with previous studies. The microstructure of the left AFFA showed a positive association with alcohol dependence risk. Previously, brain white matter microstructure was shown to be negatively associated with alcohol consumption [14,15], and consistent with the well-known toxicity of alcohol to the whole brain [5]. The left arcuate fasciculus in this study showed exceptional characteristics in which its microstructure and was positively associated with alcohol dependence risk.

Although the left AFFA microstructure showed a positive association with alcohol dependence risk, the mediation analysis revealed that this positive association was mediated by higher IQ, rather than being a direct association. The positive association between IQ and alcohol consumption is well established [2,3,4]. Stronger left AFFA is associated with alcohol dependence risk through greater IQ.

Our results need to be carefully interpreted since we have no direct knowledge about causality though it may appear that greater left AFFA causes higher IQ which causes alcohol dependence risk. Instead, we are detecting a tendency for alcohol dependence risk to be more common with greater left AFFA through its tendency for higher IQ.

Age may have a ineligible influence on the association between AFFA and alcohol dependence risk. In order to test the significance of the effect of age, additional multiple linear regressions were calculated to predict alcohol dependence risk based on left AFFA and age, for the group at the median age (45 years old) and younger, as well as for the older group. The younger group maintained a significant association but the association was not significant in the older group. It suggests the possibility that the influence of left AFFA is more profound in a younger population.

The current study was motivated by anecdotal reports from clinicians that the vast majority of clients who struggle with Alcohol Use Disorder (AUD) are quite intelligent, which contrasts with the current understanding that heavy alcohol use often negatively affects cognitive functioning. For instance, Ioime et al. [20] found functional deficits in verbal memory, visuospatial abilities, and executive functions. Widely accepted clinical implications of the harm to white matter by alcohol misuse are increased impulsive behavior and impairments in the response inhibition [21]. In worst-case situations, chronic alcohol use can result in Wernicke’s encephalopathy, Korsakoff’s syndrome, and alcohol-related dementia, all of which are permanent forms of brain damage [22]. From a neuroimaging standpoint, brain white matter microstructure was previously shown to be negatively associated with alcohol consumption [14,15].

The current study’s findings are intriguing in light of Mulhauser and colleagues’ findings that the language domain was less affected by heavy alcohol users than other domains [22]. Of further interest is the effectiveness of Acceptance and Commitment Therapy (ACT) [23] in treating AUD [24,25]. A central component of ACT is a focus on language/“mind” as a major source of clinical distress [26]. If the AFFA, heavily influential in language, is indeed associated with AUD through greater IQ, then this information has the potential for assisting the design of clinical interventions.

Clinicians need to know how cognitive functioning is affected to plan clinical interventions. It has been said that “the literature is unanimous in considering that cognitive impairment reduces effectiveness of psychological treatments” [20] and that a better understanding of cognitive functioning is necessary to plan individualized treatment. While many researchers state the need for assessing both deficits and strengths, most studies focus only on cognitive impairment [20,22]. However, there is a long-standing clinical awareness of the importance of knowing what is not impaired (client strengths) when engaging in assessment and treatment planning. Indeed, studies have shown that clinicians who are aware of the neuropsychological status of their clients not only provide better care but also have better treatment outcomes [22,27,28]. The importance of awareness of clients’ cognitive strengths is also emerging in other fields such as the treatment of psychosis [29].

## 5. Conclusions

The current study’s findings revealed that although the left AFFA microstructure showed a positive association with alcohol dependence risk, this positive association is mediated by higher IQ, rather than the left AFFA itself being directly associated with alcohol dependence risk. If the AFFA, heavily influential in language and human sentience, is indeed associated with AUD through greater IQ, this information has the potential to be quite useful when designing clinical interventions. A possibility is suggested that assessment of the left AF may provide helpful information in the future treatment of AUD.

## Figures and Tables

**Figure 1 brainsci-13-00129-f001:**
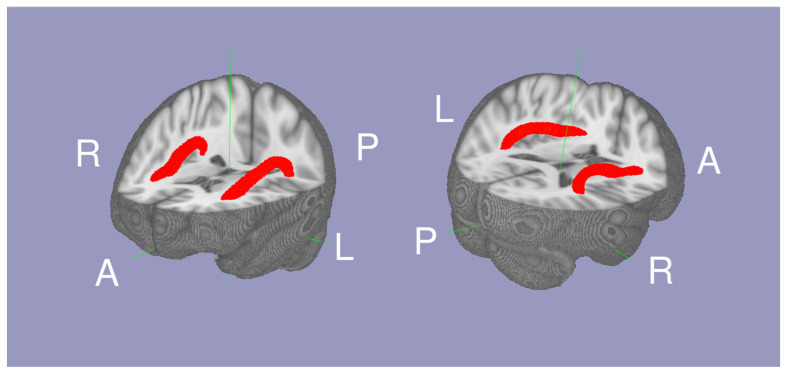
The mean probabilistic tractography of the arcuate fasciculi (red) between Wernicke’s area and Broca’s area white matter.

**Figure 2 brainsci-13-00129-f002:**
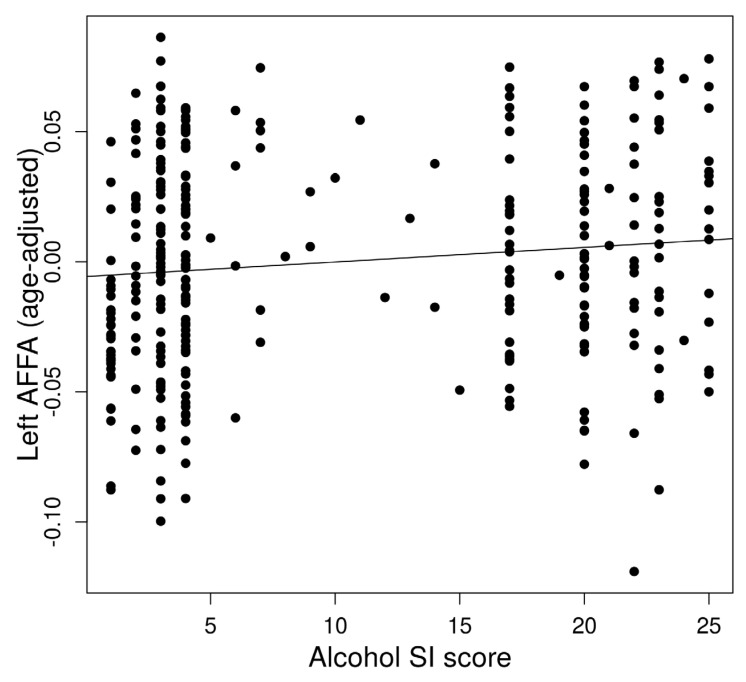
Association between alcohol dependence risk and left Arcuate Fasciculus FA (AFFA).

**Figure 3 brainsci-13-00129-f003:**
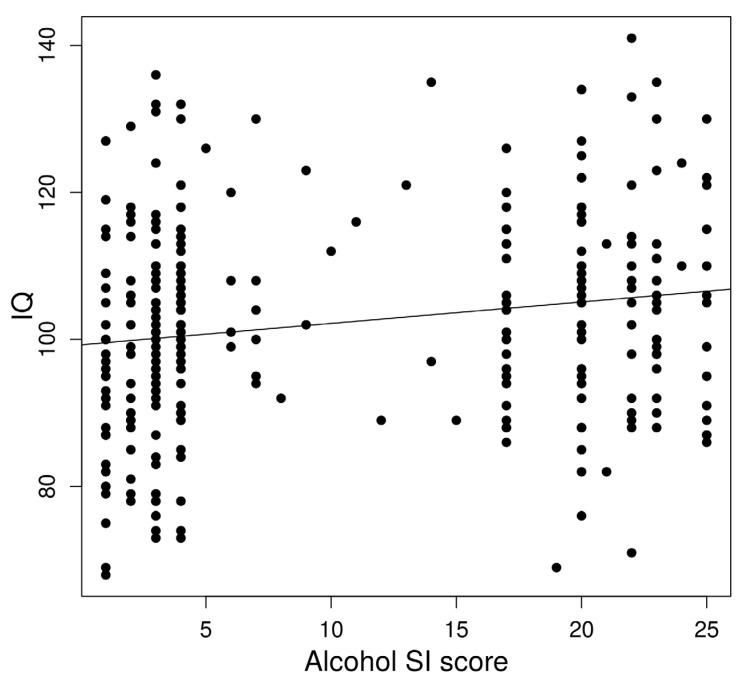
Association between alcohol dependence risk and IQ.

**Figure 4 brainsci-13-00129-f004:**
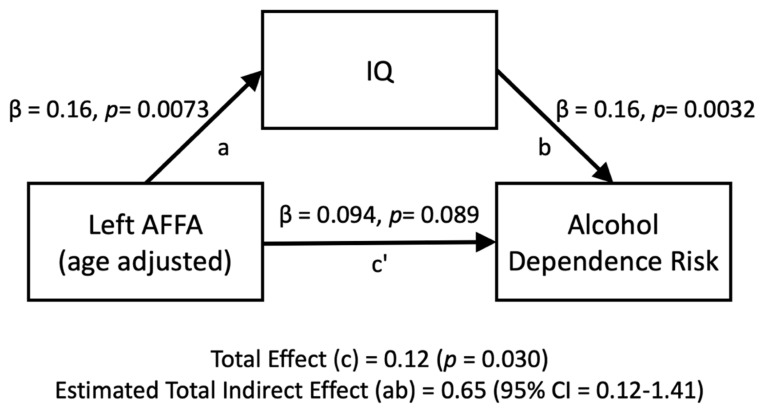
Mediation analyses for the left AFFA (age-adjusted) on the number of alcohol dependence risk, directly (c’) and indirectly through IQ (ab).

**Table 1 brainsci-13-00129-t001:** Subject Demographics.

*N*	334
Age	42.99 ± 22.78
Female:Male	221:113
IQ	102.23 ± 14.16

## Data Availability

The original data is available on the NKI-RS website. The tractography code will be available at https://olemiss.edu/projects/dnl/.

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
