# Peer review of "Arcuate Fasciculus Microstructure Predicts Alcohol Dependence Risk through Higher IQ"

_brainsci, 2023, doi:10.3390/brainsci13010129_

Round 1

Reviewer 1 Report

This interesting study examined the relationship of higher IQ, alcohol dependence and AFFA microstructure finding that the left, but not right AFFA, age and IQ predicted AUD. Unfortunately, the age range of the male/female subjects was considerable (range 16-85: 42.99 +/- 22.78). You may want to consider a follow-up analyses of a younger vs older group. l would also consider a more in depth discussion as to the exact role the AFFA might play in alcohol dependence. We know, for example, that multiple brain structure are related to AUD and it would be an excellent contribution to elucidate the AFFA's role. All in all, this was a good first step but it is hoped that the authors will continue studying the complex relationship between the AFFA microstructure, intelligence and AUD.

Author Response

Thank you very much for your suggestion. We added the younger vs older age comparison. Interestingly, the effect was only found in the younger group. We added this in the discussion section. We highly appreciate this input from you.

Reviewer 2 Report

The comments are listed below. Please address these and make the discussion a compelling one.

Figure1: AF is red, not green. Please correct it. Is it a tract density image?

Materials and methods section

Lines 64–67: Please mention the b value of DTI data here instead of mentioning it in line 74 . 

Line 89: Please abbreviate AFFA and NIDA as they are first appearing in the text. 

How long had these patients been drinking? Please provide a table showing participant characteristics such as age, gender, duration of alcohol consumption, education, etc.

Discussion

Second paragraph: The lines "The microstructure of the left AFFA showed a positive association with alcohol dependence risk" and "The left arcuate fasciculus in this study showed exceptional characteristics in which its microstructure was positively associated with alcohol dependence risk" convey the same meaning. Please avoid repetition.

Have you tried mediation analysis with age or education or interaction between them? For example, I am just wondering if being young and having a high IQ were related to the risk of alcoholism!

The fourth paragraph: The first and last lines are similar. Please delete "Our results should not be interpreted as deterministic or causative relations."

The fifth paragraph is not required. The same story from the previous paragraphs. Please delete it. 

Line 157: "AUD," please abbreviate.

The current study is motivated by anecdotal evidence from clinicians. The findings are very intriguing! In general, alcohol consumption changes speech production, making people talk too much or exhibit slurred speech. AF, being a language tract has a role. The discussion is not clear. It is noteworthy that previous studies have also shown that a lower IQ is associated with higher alcohol consumption. Moreover, besides AF, other white matter tracts such as ILF, CC, IFOF are involved in IQ. From this study, it is not clear what the underlying mechanism is. A lot of things could be going on in the brain of an alcoholic; it's worth noting that "future studies are required to confirm whether the AFFA increase might be some kind of compensatory mechanism and may require experiments analyzing other white matter tracts."

It is not clear how higher FA in the AF is helping the treatment plan. A well-trained neuropsychiatrist or neuropsychologist can reveal the cognitive abilities of AUD patients with their rating scales and suggest rehabilitation or cognitive exercise plans. How does the AFFA help them? So, you get a DTI done for your patients, track the AF, and then what information are you giving to the clinician? Are these findings patient-specific?

Here are my two cents!

The study suggests that microstructural integrity in the left AF may constitute a neuroanatomical substrate of IQ-mediated alcoholic dependence. Perhaps the authors could gently suggest that, in the future, DTI-AF tractography may aid in the selection of treatment strategies. Furthermore, using longitudinal studies, it will be interesting to see how the AFFA trajectory changes in these participants to uncover the underlying mechanisms.

Author Response

The comments are listed below. Please address these and make the discussion a compelling one.

We appreciate your input. We made a revision accordingly. Thank you very much for helping us improve this manuscript.

Figure1: AF is red, not green. Please correct it. Is it a tract density image?

Materials and methods section

Lines 64–67: Please mention the b value of DTI data here instead of mentioning it in line 74 . 

Line 89: Please abbreviate AFFA and NIDA as they are first appearing in the text. 

We made corrections to these four points above. Thank you very much for pointing them out.

How long had these patients been drinking? Please provide a table showing participant characteristics such as age, gender, duration of alcohol consumption, education, etc.

Although we don’t have any data about how long they have been drinking, we added a table to summarize participant characteristics.

Discussion

Second paragraph: The lines "The microstructure of the left AFFA showed a positive association with alcohol dependence risk" and "The left arcuate fasciculus in this study showed exceptional characteristics in which its microstructure was positively associated with alcohol dependence risk" convey the same meaning. Please avoid repetition.

This is indeed not a repetition. Our finding was positive, while previous ones reported negative associations.

Have you tried mediation analysis with age or education or interaction between them? For example, I am just wondering if being young and having a high IQ were related to the risk of alcoholism!

We did not try this but the reviewer 1 suggested us to try sub-group analysis for younger and older groups. It turned out that your guess is correct that this effect is found in the younger population.

The fourth paragraph: The first and last lines are similar. Please delete "Our results should not be interpreted as deterministic or causative relations."

We deleted this sentence.

The fifth paragraph is not required. The same story from the previous paragraphs. Please delete it. 

We deleted this paragraph.

Line 157: "AUD," please abbreviate.

We fixed this.

The current study is motivated by anecdotal evidence from clinicians. The findings are very intriguing! In general, alcohol consumption changes speech production, making people talk too much or exhibit slurred speech. AF, being a language tract has a role. The discussion is not clear. It is noteworthy that previous studies have also shown that a lower IQ is associated with higher alcohol consumption. Moreover, besides AF, other white matter tracts such as ILF, CC, IFOF are involved in IQ. From this study, it is not clear what the underlying mechanism is. A lot of things could be going on in the brain of an alcoholic; it's worth noting that "future studies are required to confirm whether the AFFA increase might be some kind of compensatory mechanism and may require experiments analyzing other white matter tracts."

As we are using brain MRI data, we are not able to determine the mechanisms. We are also interested in the underlying mechanism for these relationships.

We highly appreciate your input to improve this manuscript. Thank you very much for your valuable comments.

It is not clear how higher FA in the AF is helping the treatment plan. A well-trained neuropsychiatrist or neuropsychologist can reveal the cognitive abilities of AUD patients with their rating scales and suggest rehabilitation or cognitive exercise plans. How does the AFFA help them? So, you get a DTI done for your patients, track the AF, and then what information are you giving to the clinician? Are these findings patient-specific?

As this may be the first study that showed a potential association between AFFA and AUD, we would not be able to make clinical recommendations. Since we would not be able to modify AF or any other brain structures, we may not be able to target the AF. We believe that our study can contribute more directly to the understanding of AUD, rather than actual clinical practice.

Here are my two cents!

The study suggests that microstructural integrity in the left AF may constitute a neuroanatomical substrate of IQ-mediated alcoholic dependence. Perhaps the authors could gently suggest that, in the future, DTI-AF tractography may aid in the selection of treatment strategies. Furthermore, using longitudinal studies, it will be interesting to see how the AFFA trajectory changes in these participants to uncover the underlying mechanisms.

We highly appreciate your suggestion as well as very encouraging comments. We added a very last sentence to suggest this possibility.

Round 2

Reviewer 2 Report

Thank you for addressing most of my suggestions.